# Prediction of Dielectric Constant in Series of Polymers by Quantitative Structure-Property Relationship (QSPR)

**DOI:** 10.3390/polym16192731

**Published:** 2024-09-26

**Authors:** Estefania Ascencio-Medina, Shan He, Amirreza Daghighi, Kweeni Iduoku, Gerardo M. Casanola-Martin, Sonia Arrasate, Humberto González-Díaz, Bakhtiyor Rasulev

**Affiliations:** 1Department of Coatings and Polymeric Materials, North Dakota State University, Fargo, ND 58102, USA; estefania.ascencio@ndsu.edu (E.A.-M.); shan.he.1@ndus.edu (S.H.); amirreza.daghighi@ndsu.edu (A.D.); kweeni.iduoku@ndsu.edu (K.I.); gerardo.casanolamart@ndsu.edu (G.M.C.-M.); 2IKERDATA S.L., ZITEK, University of the Basque Country (UPV/EHU), Rectorate Building, 48940 Bilbao, Biscay, Spain; 3Department of Organic and Inorganic Chemistry, Faculty of Science and Technology, University of the Basque Country (UPV/EHU), P.O. Box 644, 48940 Bilbao, Biscay, Spain; sonia.arrasate@ehu.eus (S.A.); humberto.gonzalezdiaz@ehu.eus (H.G.-D.); 4Biomedical Engineering Program, North Dakota State University, Fargo, ND 58105, USA; 5IKERBASQUE, Basque Foundation for Science, 48011 Bilbao, Biscay, Spain

**Keywords:** dielectric constant, polymers, QSPR, Gradient Boosting Regressor, Accumulated Local Effect

## Abstract

This work is devoted to the investigation of dielectric permittivity which is influenced by electronic, ionic, and dipolar polarization mechanisms, contributing to the material’s capacity to store electrical energy. In this study, an extended dataset of 86 polymers was analyzed, and two quantitative structure–property relationship (QSPR) models were developed to predict dielectric permittivity. From an initial set of 1273 descriptors, the most relevant ones were selected using a genetic algorithm, and machine learning models were built using the Gradient Boosting Regressor (GBR). In contrast to Multiple Linear Regression (MLR)- and Partial Least Squares (PLS)-based models, the gradient boosting models excel in handling nonlinear relationships and multicollinearity, iteratively optimizing decision trees to improve accuracy without overfitting. The developed GBR models showed high *R^2^* coefficients of 0.938 and 0.822, for the training and test sets, respectively. An Accumulated Local Effect (ALE) technique was applied to assess the relationship between the selected descriptors—eight for the GB_A model and six for the GB_B model, and their impact on target property. ALE analysis revealed that descriptors such as TDB09m had a strong positive effect on permittivity, while MLOGP2 showed a negative effect. These results highlight the effectiveness of the GBR approach in predicting the dielectric properties of polymers, offering improved accuracy and interpretability.

## 1. Introduction

Dielectric permittivity is a fundamental electrical property that characterizes a material’s response when subjected to an electric field [1]. This property is related to the dielectric constant (ε) and reflects the material’s ability to align and orient electric dipoles within its structure in response to an external electric field. The greater the polarizability of the molecules, the higher the value of (ε) [2]. This property is influenced by several polarization mechanisms. In electronic polarization, the electric field distorts the electron cloud around atoms, generating temporary dipoles [3]. In ionic polarization, the electric field slightly displaces ions from their equilibrium positions in ionic materials [4]. Lastly, dipolar polarization occurs in materials with permanent dipoles, where the electric field aligns these dipoles, increasing permittivity based on molecular polarizability [4]. Even in materials such as liquids and gases that lack permanent dipoles, dielectric permittivity exists [5], as electrons or ions within the material can still shift in response to an external field, contributing to the material’s dielectric properties [3,4]. Together, these mechanisms enhance the material’s overall capacity to store electrical energy, which is reflected in the value of the dielectric constant [6,7]. This property is a fundamental characteristic of a material and commonly used to predict other electrical properties of polymers [6,7,8]. It applies to materials physics, chemistry, electrical engineering, and polymer science [1]. The implementations of this characteristic property knowledge are evident in high-energy density capacitors [9], high-voltage cables [9], microelectronics [10], and photovoltaic devices [11,12].

However, the theoretical prediction of the dielectric constant in polymers presents a multifaceted challenge. This inherently nonlinear property requires considering various factors, such as temperature, electric field frequency, polymer structure, composition, sample morphology, impurities, loads, plasticizers, and other additives [7,13]. Furthermore, each application requires the polymer’s dielectric constant (ε) to be within a specific range that meets the particular demands of that application [8]. Understanding and adjusting this range is crucial for the effective design of new materials.

Therefore, given the inherent complexity of many substances, there is a significant demand for machine learning (ML) models to efficiently predict these properties, optimizing both time and resources. In the field of materials informatics and cheminformatics, the Quantitative Structure–Property Relationship (QSPR) methodology stands out as an important machine learning-based approach. This methodology relies on machine learning models to forecast or elucidate chemical compound’s properties by leveraging distinct chemical descriptors [14]. The efficacy of the model’s predictions and its capacity to unveil the relationships between a material’s molecular or other microscopic physical properties and the targeted properties being modeled are significantly influenced by the careful selection of descriptors [14]. In this sense, the QSPR approach has proven to be effective in predicting various properties, including glass transition (Tg) in polymers [15,16] and (Tg) in polymer coating materials [17]. Several QSPR models have also been developed for predicting dielectric permittivity in polymers [1,17,18,19] using different datasets, feature-representation methods, variable selection procedures, and so on. For instance, Liu et al. [20] developed a QSPR model to predict dielectric permittivity using a small dataset of 22 polyalkenes. The resulting model utilized a multiple linear regression analysis (MLRA), had a high (*R^2^*_train_) value of 0.907, and standard error (s) of 0.001 for the training set. Three quantum-chemical descriptors were selected: ELUM (energy of the lowest unoccupied molecular orbital), q- (minimum negative atomic charge) and S (configurational entropy of the system). The authors thoroughly explored the physical significance of these descriptors, linking them to polymer polarizability and charge separation capability.

In subsequent studies in 2016, Wu et al. [19] developed a model to predict the dielectric constant based on 58 polymers. The authors employed Partial Least Squares (PLS) regression as the modeling technique, incorporating the Infinite Chain Descriptors (ICD) 2D, TAE and GAP_inf3_inv. The model trained on the training dataset showed (*R^2^*_train_) of 0.91 and a Root-Mean-Square Error (*RMSE*) of 0.11. Additionally, when evaluating the model on an external test set, it showed high *R^2^* values and achieved strong predictive capabilities, reaching an (*R^2^*_test_) of 0.96 and an *RMSE* of 0.11 in both cases. Finally, in a recent study, Zhuravskyi et al. [1] used a dataset of 71 polymer samples. The authors applied a combined genetic algorithm (GA) and multiple linear regression analysis (MLRA) to select optimal descriptors and develop predictive models. Two models were created—the first model used five descriptors, achieving an (*R^2^*_train_) of 0.842 and a standard error (s) of 0.187. The second model incorporated eight descriptors, demonstrating improved results with (*R^2^*_train_) of 0.905 and s of 0.151. Both models exhibited robust predictive skills when externally validated, showing (*R^2^*_test_) of 0.829 and 0.810 for training and test sets, respectively.

Although all of these earlier publications report on QSAR/QSPR studies to predict dielectric permittivity of different polymers, they all have certain limitations. First of all, not all models use separated sets for training and test sets to validate the models’ predictions; as well, the size of published datasets is smaller and/or limited in comparison to the current model, restricting the applicability domain of the previous models. 

Additionally, methods like Multiple Linear Regression (MLR) are vulnerable to multicollinearity, leading to unstable coefficients and overfitting, as well as an increased risk of identifying misleading relationships between variables, especially when many variables are involved [21]. However, Partial Least Squares (PLS) handles multicollinearity well, but it may overlook important relationships by focusing primarily on general trends. Moreover, its accuracy can be compromised if the variables are on very different scales, complicating model interpretation [21,22]. Also, nonlinear correlations may not be well captured by these linear methods, limiting their ability to model complex relationships accurately [23]. In contrast, gradient boosting (GB) models are highly effective at managing both multicollinearity and nonlinear relationships between variables [24]. The method is very powerful, since it is updating the weights after each iteration, influencing precise models in the sequence for continuous improvement of overall accuracy over time [25,26]. Thus, GB has been successfully used in QSAR models to predict bandgap [27] and glass transition temperature [28] in polymers, with predictive capability of *R^2^*_train_ above 0.90 in both cases, where high prediction quality was achieved even with many descriptors without overfitting [29].

In this work, a QSAR model was developed using a dataset of 86 polymers. Two versions of the model (GB_A and GB_B) were evaluated using cross-validation and external datasets. The optimization of the models involved the use of eight descriptors, and six descriptors, respectively. Parameters of the Gradient Boosting model, such as criterion, max_features, min_samples_leaf, max_leaf_nodes, and min_impurity_decrease [30], were optimized using the grid search technique [24]. These hyperparameters (Table 1) are crucial for enhancing model accuracy [31], significantly increasing the model’s ability to capture complex relationships between input and output variables, prevent overfitting, and ensure robust decision-making [31,32]. The optimized model demonstrated an effective prediction of the dielectric constant in various types of polymers.

Also, in this study the Accumulated Local Effect (ALE) approach was used to facilitate the visualization of the individual impact of each descriptor on dielectric permittivity predictions. ALE graphs serve as effective tools for both visualizing and quantifying the individual influence of each input on prediction [33]. Although, several interpretative methods exist, such as Partial Dependence Plots (PDP) and Individual Conditional Expectation (ICE) curves, which have been used in various studies [34,35,36,37]. ALE plots offer more precise interpretations in complex models. The ALE plots do so by mitigating inaccuracies caused by the aggregation of heterogeneous effects and incorrect assumptions of feature independence [34,37]. Moreover, ALE plots allow for the identification of precise local effects within the data, thereby improving the understanding of variable interactions—something that ICE curves and PDPs are less effective at achieving. Additionally, ALE plots are more computationally efficient, overcoming the limitations of PDPs in high-complexity scenarios [36].

To our best knowledge, to date only one study has utilized the ALE method to elucidate the mechanistic relationship of nonlinear QSAR models related to toxicity (log LD50) discussed in work [33]. However, no previous studies have been identified that apply this approach to investigate dielectric permittivity.

## 2. Materials and Methods

### 2.1. Experimental Data Collection

In this study, we examined a set of 86 polymers (Appendix A) compiled from diverse public sources [1,7,38,39]. The dataset encompasses various polymer types, including polyvinyls, polyethylenes, polyoxides, polystyrenes, polyethers, polysulfones, polyacrylonitrile, polyamides, polyacrylates, poly-siloxanes, polyxylylenes, polycarbonates, polyisoprenes, polymethylene, aromatic polymers, fluorinated polymers and norbornene polymer. 

All experiments were conducted at a temperature of 298 K, with measurements taken at frequencies of 1, 60, 100, 1000, 10,000, and 1,000,000 Hz. To ensure data consistency, the dielectric constant values obtained at these frequencies were extrapolated to 1 Hz using linear regression equations (Appendix A). This allowed for a coherent comparison of dielectric permittivity under the same frequency conditions. The quality of the fits was guaranteed by a coefficient of determination (*R^2^*) greater than 0.90.

The SMILES linear notation system was used for each polymer, representing molecular structures in a compact text format, which makes it useful for chemistry software and data exchange [40], and the SMILES notations for each polymer were obtained from PubChem [41] and ChemDraw [42]. The molecules were optimized by Avogadro Software version 1.2.0. [43] with Universal Force Fields (UFFs). A UFF is a general force field designed to optimize minimal energy conformation that works for chemical structures based on all possible elements. It determines parameters based on the element, its hybridization, and connectivity; this force field is a big advantage over other force fields that usually only work in specific cases depending on the available parameters [44]. 

### 2.2. Generation of Descriptors

Molecular descriptors are mathematical representations of the molecular properties generated by specific algorithms based on mathematical equations [45]. The descriptors were generated using alvaDesc [46]. The program calculates more than 5000 descriptors, 0-dimensional, 1-dimensional, 2-dimensional, and 3-dimensional, GETAWAY descriptors, among others [46]. Highly correlated descriptors (R > 0.9), constant, and near-constant descriptors (std < 0.1) were removed during pre-processing. After eliminating correlated, constant, and near-constant descriptors, about 1273 descriptors were used for further QSPR analysis.

Additionally, given the higher molecular weight of the polymers, the influence of terminal groups on the overall polymer structure is minimal. Consequently, we can disregard the contribution of terminal structures. In this context, we based our calculations of structural features/descriptors on the repeating polymer units’ structures [1,20,47].

### 2.3. Model Assembly

For model construction and QSPR evaluation, the dataset was organized in ascending order by the target property and split into an 80% training set and a 20% test set. The preliminary phase, illustrated in the data distribution, consists of identifying and excluding 4 atypical structures from the dataset using a histogram [48]. Subsequently, a lower limit (lower limit) was established by subtracting three times the standard deviation (σ) of the mean (χ): Lower Limit=χ−3xσ and an upper limit (upper limit) by adding three times the standard deviation of the mean: Lower Limit=χ+3xσ. This approach was in line with the empirical standard in a normal distribution [48]. Several models, including Multilinear Regressor (MLR), Support Vector Machine (SVM), Random Forests (RFR), Decision Tree (DT), K-Nearest Neighbors (KNN), and Gradient Boosting (GB) were built for further evaluation and identification of the best model. These models were generated with the coefficient of determination in the training dataset (*R^2^*_train_) and the validation dataset (*R^2^*_test_) parameters. Model acquisition was performed using Python (3.6.3) and implemented in the Scikit-learn package [49]. The selection of variables was made with Genetic Algorithm (GA), a robust tool for search and optimization in predictive modeling [50,51]. The variable selection process using Genetic Algorithms (GA) begins with an initial population of 1000 random models. The evolutionary phase involved 9000 iterations, and a mutation probability of 20% was applied.

### 2.4. Gradient Boosting Regressor Model Modeling and Validation 

The Gradient Boosting Regressor model used several performance metrics, including the coefficient of determination (*R^2^*) Equation (1), Root-Mean-Square Error (*RMSE*) (Equation (2), and Mean Absolute Error (*MAE*) Equation (3). These metrics are commonly employed to evaluate the effectiveness of the model [1,14,33]. In this context, $y_{i}^{obs}$ and $y_{i}^{pred}$ refer to the actual and predicted values for each compound, while ${\tilde{y}}_{i}^{obs}$ is the average of the observed values. In this particular case, each ith compound is characterized by only one observed value. To assess the model’s stability, we computed the Mean Absolute Error of cross-validation (*MAECV*) in each iteration based on Equation (4). Similarly, we used the Concordance Correlation Coefficient (*CCC*, Equation (5)) to gauge the goodness of fit. Additionally, other metrics were incorporated to obtain a more comprehensive and precise estimation of the models’ predictive capacity. The external predictability of the model was assessed using metrics such as Q^2^F1, Q^2^F2, k, k’ [52].
(1)$$R^{2}=1-\frac{\sum_{i=1}^{n}{\left(y_{i}^{obs}-y_{i}^{pred}\right)}^{2}}{\sum_{i=1}^{n}{\left(y_{i}^{obs}-{\tilde{y}}_{i}^{obs}\right)}^{2}}$$
(2)$$RMSE=\frac{\sqrt{\sum_{i=1}^{n}{\left(y_{i}^{obs}-y_{i}^{pred}\right)}^{2}}}{n}$$
(3)$$MAE=\frac{1}{n}\overset{n}{\underset{i=1}{\sum }}\left|y_{i}^{obs}-y_{i}^{pred}\right|$$
(4)$$MAECV=\;\;\;\;\frac{1}{n}\;\;\overset{n}{\underset{k=1}{\sum }}\left[\;y_{i}^{obs}-y_{i}^{predcv}\right]$$
(5)$$CCC=\frac{2\sum_{i=1}^{N\;obs}\left(y_{i}^{obs}-y^{-obs}\right)\left(y_{i}^{pred}-y^{-pred}\right)\;}{\sum_{i=1}^{n\;}{\left(y_{i}^{obs}-y_{i}^{-obs}\right)}^{2}+\;\sum_{i=1}^{n}{\left(y_{i}^{pred}-y^{-pred}\right)}^{2}+n{\left(y^{-obs}-y^{-pred}\right)}^{2}\;}$$

### 2.5. Analysis of Descriptors in Models

The research attempts at overcoming the challenge of interpreting nonlinear models by employing the ALE approach. This approach proves effective in comprehending the impact of descriptors on the target variable [33,53]. Additionally, data normalization was performed to ensure consistency in interpreting ALE effects, thereby ensuring a precise understanding of how each descriptor (Table 2) influences the model’s predictions. The Scikit-learn package [49] was utilized for normalizing the descriptors, and ALE Python package [33] to generate graphical representations that visualize the cumulative effects of the descriptors on the predictions of the GB_A and GB_B models.

## 3. Results and Discussion

### 3.1. Exploratory Data Analysis

Data visualization through histograms is a fundamental step in quantitative data analysis [54]. In this study, histograms were used to illustrate the distribution of dielectric permittivity in the dataset. Figure 1A shows the original dataset of 86 polymers. The *X*-axis represents dielectric permittivity values, while the *Y*-axis shows the frequency of each value or range. The taller the bar in the histogram, the more frequently those values appear in the dataset [55]. Additionally, the blue lines represent density curves generated using Kernel Density Estimation (KDE), which provide a smooth and continuous view of the data distribution. These curves help highlight concentrations and central trends in the data [56]. In addition, Figure 1 reveals that most data points cluster around the center, with fewer at the extremes, showing a right-skewed distribution [51]. The mean value (3.148) serves as the central point, with a lower limit of −0.546 and an upper limit of 6.844, calculated by subtracting and adding three times the standard deviation (1.232), respectively. Data points beyond these limits were flagged as outliers, including Fumaronitrile (8.5), Vinyl Fluoride (8.5), Vinylidene Fluoride (8.4), and Methylcellulose (6.8). After this step, the dataset was reduced to 82 points. The updated graph (Figure 1B) shows a significant improvement in the accuracy and reliability of the data analysis, ensuring a more precise representation of the dataset and achieving a normal distribution.

Fitting data to a distribution curve through histograms is crucial for identifying general patterns and detecting outliers that may influence the results, ultimately providing deeper insight into the data’s behavior and trends [56].

### 3.2. Ensemble Model

After an initial preprocessing phase, a dataset of 82 polymers were split into training and test datasets, containing 66 and 16 polymers, respectively. A total of 1273 descriptors were generated for this dataset. Using these descriptors, various models were developed using the following algorithms: Multi-Linear Regressor (MLR), Support Vector Machine (SVM), Random Forests (RF), Decision Tree (DTR), K-Nearest Neighbors (KNN), and Gradient Boosting (GB) (Figure 2). When analyzing several models for the predictive performance, the coefficient of determination (*R^2^*) was assessed, aiming for the coefficient’s value close to 1 [52]. Such models as MLR, SVM, and DTR achieved values close to 0.6. However, models like RF demonstrated high training performance with values near 0.9, but their validation performance significantly dropped to around 0.6. Similarly, the K-NN model showed consistent performance in both training and validation sets, with values close to 0.65 (Figure 2). Nevertheless, the GB models proved to be effective in predicting dielectric permittivity, surpassing the other ML models. Two options were chosen that performed better for the Gradient Boosting (GB) model. The first model (GB_A) consisted of eight descriptors, and the second model (GB_B) consisted of six descriptors (Table 2). Additionally, a hyperparameter optimization was conducted for each model (Table 1). This optimization was crucial to significantly improve the predictive performance of the model while reducing the risk of overfitting by simplifying its complexity [57]. Statistical parameters of the model are presented in Table 3.

In the model, GB_A (*R^2^*_train_) shows a good performance, indicating the model’s ability to capture and explain 93.77% of the variations in the training data, showcasing its effective adaptation and precise predictions within this specific set. As for the test set, the *R^2^*_test_ value of 0.801 illustrates a very good model’s performance on the external set, which was not used during the model training. This high performance in both training and test sets highlights the model’s robustness, supporting its ability to generalize and provide accurate predictions in future scenarios. In contrast, the GB_B model showed slightly lower prediction ability for both the training set (*R^2^*_train_): 0.822 and the test set (*R^2^*_test_) 0.708. Therefore, we could assume that having more descriptors might allow the model to capture more details and subtle relationships in the data, potentially improving the accuracy of predictions; however, this improvement could introduce a higher complexity to the model. In Figure 3, the relationship between predicted and experimental values for the dielectric constant is illustrated, comparing models GB_A (Figure 3A) and GB_B (Figure 3B). It can be observed that residual errors are small for model GB_A, in contrast to model GB_B. Additionally, the black line represents the regression line associated with the data points, where residual errors are evident.

Without prejudice to the statistics discussed above, other important performance parameters that should also be considered for the selection of good predictive models are the mean-root-quadratic error (*RMSE*) and Mean Absolute Error (*MAE*) [52]. In our study, when comparing the GB_A and GB_B models, the highest *R^2^* values for training and test sets were consistently correlated with lower *RMSE* and *MAE* values. Thus, model GB_A highlights the ability of the model to make better predictions that are quite close to real values. Furthermore, the data were assessed in a cross-validation set (CV) for Models A and B, resulting in a similar *MAECV* of 0.2605 and 0.2725, respectively. This indicates that the models’ predictions during cross-validation have an average absolute error of around 0.27 units compared to the actual values.

Previous research [1] reported a QSPR model that utilized GA-MLR analysis. The models developed were generated from 71 polymers, achieving an (*R^2^*_train_) of 0.905 on the training set and an external (*R^2^*_test_) of 0.812 on the test set. This dataset served as the main starting point for our study, to which an additional set of polymers was added. It is crucial to highlight that despite of our study employing a gradient boosting model, the consistency of the results between this model and the one developed earlier suggests the robustness of this methodology. This approach demonstrates its effective ability to capture the relationship between predictor variables and the response variable, even when considering an expanded dataset with the inclusion of additional polymers. 

In a similar study, Bicerano [7] crafted a QSPR model achieving an (*R^2^*) of 0.958 and (s) of 0.087, aiming to establish a correlation between (ε) and 32 descriptors related to the structure of 61 polymers. However, this model’s complexity initiates from an abundance of descriptors, potentially leading to issues like overfitting. The decision to augment the descriptor count may have enhanced results, yet it introduces complexity affecting the model’s reliability. Moreover, the model in the discussed paper lacks an external validation, i.e., no test set is utilized. In a similar way, Xu et al. [18] employed a dataset comprising 57 polymers. Instead of using simple repetitive units, they utilized cyclic dimers to represent polymer structures, providing a more accurate capture of the chemical environment’s impact. In total, nine descriptors related to composition, connectivity, charges, and topological indices were selected in the model discussed. The QSPR model yielded (*R^2^*_train_) of 0.938 and standard error (s) of 0.087, using the MLRA. Furthermore, the model underwent the external validation on a test set of 12 polymers, achieving notable results with an (*R^2^*_test_) of 0.969.

While these studies have produced high predictions, the limited dataset limits polymer diversity and the scope of predictions. The gradient boosting model provides greater flexibility compared to the MLR model. This model refines an additive model by optimizing regression trees and minimizing the loss function. The “additive nature” means that the model gradually builds complexity, improving its accuracy progressively. The tree-based approach involves constructing decision trees, allowing the model to capture nonlinear relationships between input and output variables, and adept at handling complex patterns in the data. This approach offers versatility in modeling various aspects, including interactions between variables, capturing discontinuities, and effectively handling non-monotonous effects present in the dataset [25].

### 3.3. ML-QSPR Models Explanation 

Molecular descriptors are essential in cheminformatics [58], as they enable models to identify patterns that influence the dielectric permittivity of polymers [1]. By converting molecular structures into numerical values, these descriptors allow for the efficient analysis of factors such as geometry, atomic arrangement, bonding patterns, molecular size, and electronic properties [59]. This encoding helps predictive models assess how these structural factors affect dielectric permittivity [1]. According to Table 2, the GB_A model comprises eight descriptors, and the GB_B model comprises six descriptors. These models share two descriptors: TDB09m (spatial 3D molecular geometry and atomic properties of polymeric structures) [59] and MLOGP2 (squared Moriguchi octanol–water partition coefficient, a descriptor of lipophilicity indicating a molecule’s affinity for non-polar environments based on molecular characteristics such as hydrophobicity, ring structures, hydrogen bonds, etc.) [60]. MLOGP2 is described as a descriptor that could negatively influence dielectric permittivity as its values increase. In polymers such as poly(4-methyl-1-pentene), with a dielectric permittivity of 2.13 and an MLOGP2 of 12.363 (Figure 4B), its high lipophilicity reduces polarizability, thereby lowering the permittivity. In contrast, nylon 6, with an MLOGP2 of 0.315 and a permittivity of 3.50, exhibits greater polarity, which facilitates better molecular orientation under an electric field. This suggests that lower MLOGP2 values are associated with higher dielectric permittivity due to more effective polymer polarization. This descriptor could suggest that polymers based on repeating units with low MLOGP2 values (highly polar) are likely to exhibit high dielectric permittivity. This could be due to the fact that molecular chains distort and orient easily in response to an electric field. Contrary to the previous descriptor, the TDB09 descriptor has a different effect, whereby an increase in the values of this descriptor has a positive impact on dielectric permittivity. This could imply that polymers with larger structures or higher atomic mass may have a greater capacity to respond to an electric field, thus improving their dielectric performance.

Additionally, the GB_A model includes the descriptor HATS2p, and the GB_B model includes the descriptor RTs+, where both belong to the GETAWAY type; this type of descriptor is related to 3D molecular geometry using atomic weights, like atomic mass, polarizability, van der Waals volume, electronegativity, and unit weights [61]. Therefore, these descriptors could capture molecular interactions based on distances and atomic weights, directly influencing how the molecules respond to an electric field.

Among the selected descriptors in each model, the GB_A model has descriptors related to the type of constitutional indices. For example, N% (percentage of nitrogen atoms) quantifies the proportion of nitrogen atoms in the polymers of the data [59], the presence of functional groups in polymers, such as amino (-NH_2_) or cyanide (-CN), which could determine the behavior of this descriptor towards the property in a positive correlation, where more functional groups lead to higher permittivity. 

The descriptors P_VSA_e_3 and P_VSA_i_1 are directly related to the van der Waals surface area (VSA), showing a specific characteristic in a defined area [62,63], where the contribution of these interactions is influenced by Sanderson electronegativity for the first descriptor and ionization potential for the second descriptor, showing a positive trend in both cases, until the descriptor values reach their medium values. Another descriptor is J_Dz(p), which belongs to the 2D type descriptors based on topological representation. It represents a Balaban type index of the polarity-weighted Barysz matrix [59,64]. Finally, AVS Coulomb provides a measure of the mean electrostatic interactions between atoms in a 3D molecular structure, taking into account both repulsion and nuclear charge effects, which require 3D coordinates for all atoms, including hydrogen atoms [65].

The GB_B model also includes GATS2s descriptors, which are capturing the similarity between pairs of atoms in the molecule separated by a certain topological distance or lag [66]. This descriptor is related to important properties in dielectric permittivity, such as electronegativity and polarizability, and includes effects of atomic mass and volume, for fragments that have 2 or more bonds (lag2). Another descriptor, Eig08_AEA (ri), belongs to the Edge Adjacency Indices type, based on the edge adjacency matrix of a graph, providing the sum of all edge entries in the graph’s adjacency matrix [67]. Lastly, the descriptor WHALES60_Rem belongs to the WHALES type.

Figure 4 shows that the descriptors HATS2p and N% do not show a significant effect on dielectric permittivity. However, P_VSA_e_3 and P_VSA_i_1, up to values close to 0.25, have a positive influence on model predictions. For the first descriptor, the molecule Poly(2,2-(m-phenylene)-5,5-bibenzimidazole) (Figure 4B) could be involved in P_VSA_e_3 due to its high electronegativity. This molecule contains nitrogen atoms, which are highly electronegative, facilitating significant charge distribution within the polymer structure. This behavior aligns with Sanderson’s electronegativity represented by P_VSA_e_3, contributing to increased polarization, a crucial factor in enhancing dielectric permittivity in response to an electric field [7]. However, the molecule Poly (diallyl phenyl phosphonate) could be involved in the P_VSA_i_1 descriptor due to the presence of atoms like phosphorus, which impact the material’s ionization potential. P_VSA_i_1 is linked to the ionization capacity of the molecule, suggesting that compounds with such functional groups can improve the polymer’s responsiveness to an electric field, thus enhancing its polarization and dielectric permittivity [7,68,69].

The descriptor TDB09m shows similar behavior to the two previous descriptors. Polymers with heavier structures, such as Poly (bisphenol A carbonate) and Poly(1,4-cyclohexylidene dimethylene terephthalate), tend to exhibit higher dielectric permittivity due to their greater mass and structural complexity. These attributes enable better polarization in response to an electric field, thereby enhancing their energy storage capacity. In contrast, lighter polymers like Poly(propylene) and Poly(isobutylene) have less mass and simpler structures, which limit their ability to polarize effectively, resulting in lower dielectric permittivity and reduced energy storage efficiency (Figure 4B and Figure 5B).

On the other hand, the descriptor MLOGP2 shows a negative effect until values close to 0.65. As for the descriptor AVS_Coulomb, it shows a significant positive effect within values of the approximate range of 0.25 to 0.50. Finally, the descriptor J_Dz(p) also shows a subtle negative trend around the values close to 0.25; this descriptor potentially correlates with dielectric permittivity, as it captures aspects of molecular structure associated with polarity and electronic distribution [59,64]. In general, when analyzing the trends of several descriptors for the GB_A model, we could infer that values close to 0.25 can mean a remarkable turning point in how descriptors influence the prediction of dielectric permittivity.

The ALE graphs for the GB_B model provide important information on the factors influencing the dielectric constant in investigated polymers. The descriptor RTs+ does not have a significant impact on dielectric permittivity for most of the part of values, except smallest ones, close to 0. Nevertheless, for the two shared descriptors, in the two TDB09m and MLOGP2 models, it should be noted that they behave similarly, showing strong positive and negative trends, respectively. Therefore, we can conclude that these descriptors play a crucial role in determining dielectric permittivity in our models. However, in the GB_B model, the descriptor TDB09m has a more pronounced positive effect when its values increase beyond 0.65. In the same way, the descriptor WHALES60_Rem also shows a positive effect when its values are around 0.25, but for higher values, a constant behavior is observed. This descriptor belongs to the WHALES type of descriptors, and based on 3D structure considering all atoms and bonds, along with distances between them and other important properties, such as electronegativity [70].

## 4. Conclusions

In this work, two models were developed to predict the dielectric constants (ε) for various polymers providing a detailed explanation from a mechanistic perspective. The study introduced QSPR models developed by applying the Gradient Boosting algorithm. The GB_A model, having eight descriptors, showed better performance with (*R^2^*_train_) = 0.938 and (*R^2^*_test_) = 0.802, while the GB_B model, which has six descriptors, showed (*R^2^_t_*_rain_) = 0.822 and (*R^2^*_test_) = 0.704. The validity of the models was additionally ensured by various statistical verification methods, such as *MAE* and *RMSE*. The contribution of each descriptor to dielectric permittivity was discussed by applying the Accumulated Local Effect (ALE) approach. This approach worked well in analyzing the individual influence of each descriptor on dielectric permittivity predictions. Both developed QSPR-GBR models have five common descriptors that showed strong positive effects on dielectric permittivity, while one common descriptor (MLOGP2) showed a negative effect. It is important to note that TDB09m was also involved in these two models, having a positive effect. In conclusion, this study demonstrated an appropriate approach to guide the prediction of dielectric constants in a wide range of polymers, using nonlinear models. The ability to predict the dielectric constant through models, with relationship-related interpretations in ALE plots, not only optimizes the design of polymers with specific electrical properties but also accelerates the development of polymeric materials for practical applications, reducing the need for costly and lengthy experiments.

## Figures and Tables

**Figure 1 polymers-16-02731-f001:**
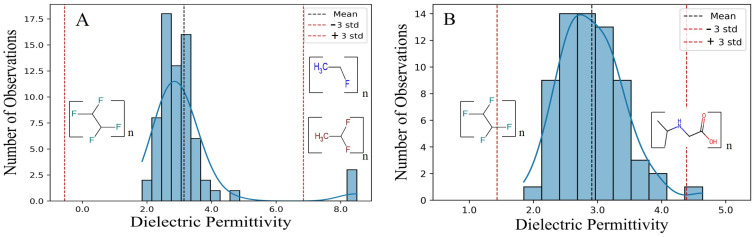
(**A**) The original dataset includes dielectric permittivity values for 86 polymers. (**B**) After removing outliers, the dataset is reduced to 82 polymers. In both histograms, the *x*-axis represents dielectric permittivity values, while the *y*-axis indicates the frequency of their appearance. The blue lines, generated using Kernel Density Estimation (KDE), illustrate the data distribution and highlight central trends.

**Figure 2 polymers-16-02731-f002:**
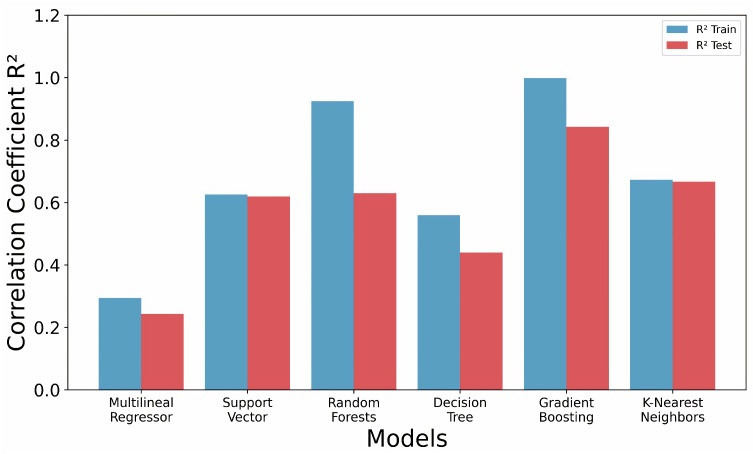
Comparison of the predictive performance of various machine learning models in estimating the dielectric permittivity of polymers. The graph displays the coefficients of determination (*R^2^*) for each model across both training and test sets.

**Figure 3 polymers-16-02731-f003:**
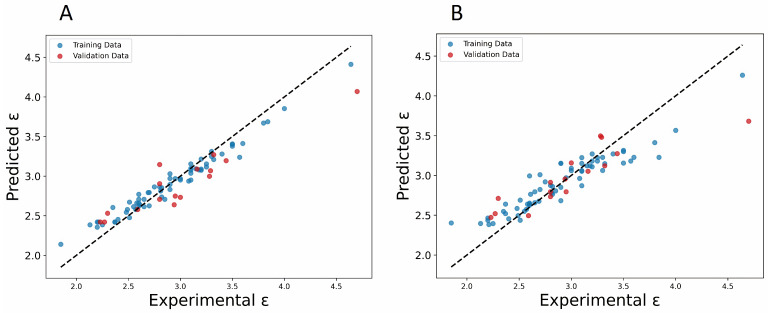
Plots of experimental vs. predicted values of the dielectric constant for (**A**) GBR_A and (**B**) GBR_B models.

**Figure 4 polymers-16-02731-f004:**
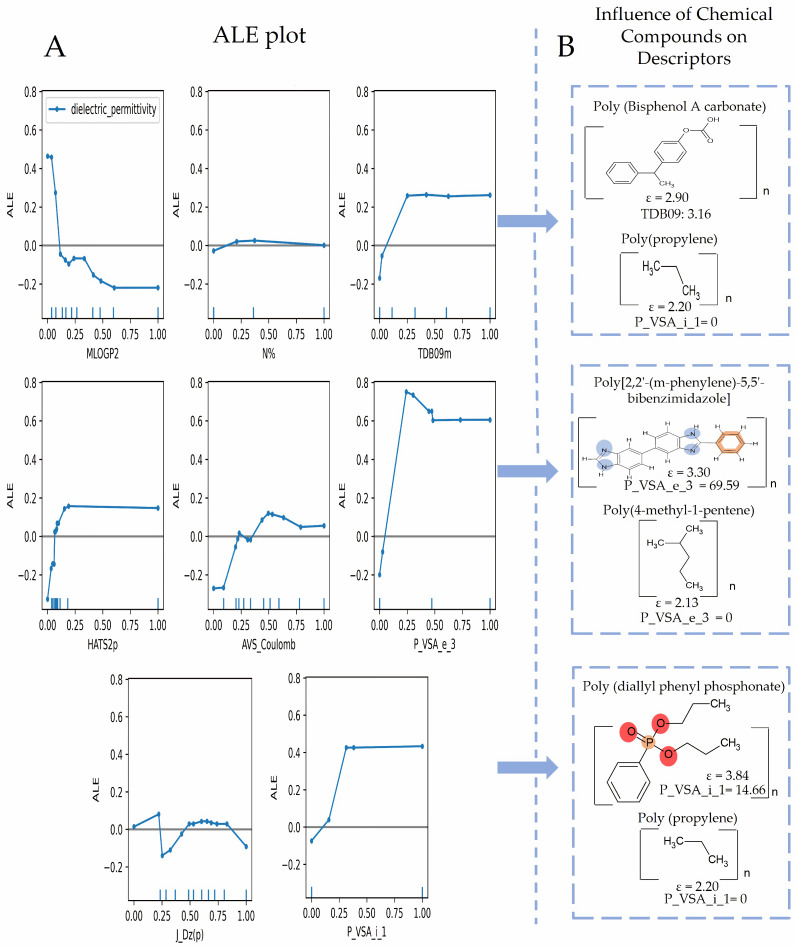
(**A**) Accumulated Local Effect (ALE) plots for the descriptors in the GB_A model, illustrating the influence of each descriptor on the prediction of dielectric permittivity. (**B**) Chemical compounds highlighting the positive or negative impact on the descriptors.

**Figure 5 polymers-16-02731-f005:**
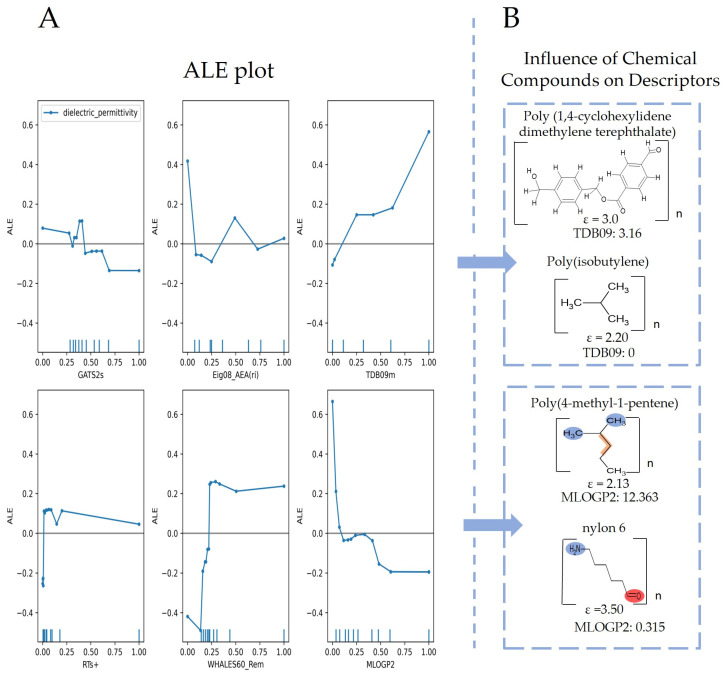
(**A**) Accumulated Local Effect (ALE) plots for the descriptors in the GB_B model, illustrating the influence of each descriptor on the prediction of dielectric permittivity. (**B**) Chemical compounds highlighting the positive or negative impact on the descriptors.

**Table 1 polymers-16-02731-t001:** Runtime parameters for Gradient Boosting Regressor.

Model Type	Common Values	Unique Values
Gradient Boosting Regressor_A	alpha: 0.9; ccp_alpha: 0.0; criterion:friedman_mse; init: None; learning_rate: 0.2; loss: squared_error;	max depth: 4; n estimators: 10
	max_features: None; max_leaf_nodes: None; min_impurity_decrease: 0.0; min_samples_leaf: 1;	
Gradient Boosting Regressor_B	min_samples_split: 2; min_weight_fraction_leaf: 0.0; n_iter_no_change: None; random_state: 42;	max depth’: 2; n estimators: 13
	subsample: 1.0; ‘tol’: 0.0001; validation_fraction: 0.1; verbose: 0; warm_start: False.	

**Table 2 polymers-16-02731-t002:** Descriptors involved in the GBR models and their corresponding definition.

Descriptor	GBR_A	GBR_B	Definition and Scope	Descriptor Type
N%	X		percentage of N atoms	Constitutional Indices
J_Dz(p)	X		Balaban-like index from Barysz matrix weighted by polarizability	2D matrix-based descriptors
P_VSA_e_3	X		P_VSA-like on Sanderson electronegativity, bin 3	P_VSA-like descriptors
P_VSA_i_1	X		P_VSA-like on ionization potential, bin 1	P_VSA-like descriptors
AVS_Coulomb	X		Average vertex sum from Coulomb matrix	3D matrix-based descriptors
TDB09m	X	X	3D Topological distance-based descriptors lag 9 weighted by mass	3D autocorrelations
HATS2p	X		leverage-weighted autocorrelation of lag 2/weighted by polarizability	GETAWAY descriptors
MLOGP2	X	X	squared Moriguchi octanol–water partition coeff. (logP^2)	Molecular properties
GATS2s		X	Geary autocorrelation of lag 2 weighted by I-state	2D autocorrelations
Eig08_AEA (ri)		X	Eigen value n. 8 from augmented edge adjacency mat. weighted by resonance integral	Edge adjacency indices
RTs+		X	R maximal index/ weighted by I-state	GETAWAY descriptors
WHALES60_Rem		X	WHALES Remoteness (Rem) (percentile 60)	WHALES descriptors

**Table 3 polymers-16-02731-t003:** Statistical parameters of Gradient Boosting model.

Model	*R^2^* (Train)	*RMSE* (Train)	*MAE* (Train)	*MAECV*	*R^2^* (Test)	*RMSE* (Test)	*MAE* (Test)	*CCC* (Test)	Q2_F1_	Q2_F2_	k	k′
GBR_A	0.938	0.123	0.100	0.261	0.802	0.256	0.212	0.869	0.805	0.802	1.035	0.961
GBR_B	0.822	0.208	0.155	0.273	0.704	0.313	0.213	0.787	0.710	0.704	0.101	0.980

## Data Availability

The original contributions presented in the study are included in the article/Appendix A, further inquiries can be directed to the corresponding author.

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
