# Peer review of "Prediction of Dielectric Constant in Series of Polymers by Quantitative Structure-Property Relationship (QSPR)"

_polymers, 2024, doi:10.3390/polym16192731_

Round 1
Reviewer 1 Report
Comments and Suggestions for Authors
The Article is devoted to the development of methods for calculating the dielectric constant of polymers using the QSPR technique based on the use of special computer programs. Similar works were carried out earlier and the starting point of the Article was the work [1]. Developing the approach proposed in [1], where a set of 71 different polymers was considered, the Authors used a wider set of 86 polymers and applied the QSPR and Gradient Boosting (GB) methods and also the Cumulative Local Effect approach to analyze the calculations. As a result, a model was developed that satisfactorily describes the value of the dielectric constant of various polymers and makes it possible to predict this value. Since the results obtained in the work are for an extended set of polymers, the article is of interest. The following comments can be made on the text of the article:
1. The description of the calculation procedure remains unclear to the reader. What parameters are included in the model – chemical composition, distances between atoms, chemical bonds? What exactly do the descriptors describe? Based on what conditions is the required number of descriptors and their type determined? The article provides graphs corresponding to various descriptors, but their connection with permittivity remains unclear.
2. The authors use the data available in the literature on the measurements of permittivity at several frequencies in the range of 60 Hz - 1 MHz and extrapolate them to a frequency of 1 Hz. It remains unclear why extrapolate to a frequency for which there are no experimental data? Note that the permittivity of polymers usually increases significantly with decreasing frequency, which indicates a difference in its mechanisms at high and low frequencies. A fitting formula that describes data in the range of 60-100,000 Hz may not work for 1 Hz. Its use must be justified, otherwise the connection with the experiment is completely lost. Figure S1 is unclear. What is shown on the horizontal axis in this figure?
Also, what frequency do the permittivity values presented in the Supplementary Materials correspond to?
3. It remains unclear what Tables 1 and 2 show. Are these tables necessary? There is no description of Tables 1 and 2 in the text.
4. The article contains statements that raise questions, stylistic errors, and the captions to the figures are not clear enough. The description of the figures in the text requires more specific explanations. These and other comments are provided in the attached pdf file.
The list of references contains references to works of recent years and is not overloaded with self-citations. The main results are confirmed by the data presented in the article.
In the opinion of the reviewer, the article needs significant revision.

Minor editing of English language required.
Reviewer 2 Report
Comments and Suggestions for Authors
The paper is interesting and well written. It discusses novel methods (machine learning models) applied to prediction of physical properties of polymers - a topic that is still being actively explored by researchers.
Three points should be addressed before publication:
1. Figure 1A does not include the outliers mentioned in the main text, contrary to the figure caption which says all 86 polymers are included.
2. Discussion of Figure 2 (section 3.2) mentions that "overall, all values fell below 0.6" which is not consistent with the actual figure which shows larger values for a few models.
3. Introduction (p.3): ALE should be expanded as 'Accumulated Local Effects', rather than 'Cumulative' (or Accumulative, as spelled in a few other places). Please use the same expansion consistently in all places.
Reviewer 3 Report
Comments and Suggestions for Authors
The manuscript presents a well-conducted study on predicting the dielectric constant of polymers using Quantitative Structure-Property Relationship (QSPR) models. The work is of high relevance to the field of materials science and cheminformatics, particularly in applications related to electrical engineering and polymer science. There are some issues should be stressed clearly before we could make the final decision.
1. In the introduction, the authors highlight the need for advanced predictive models and the relevance of QSPR approaches. However, it could benefit from a more detailed discussion on the limitations of existing methods, specifically how the proposed GBR model addresses these limitations.
2. The manuscript mentions that this is the first study to apply the ALE method to investigate dielectric permittivity. It is suggested to include a brief comparison with other interpretative methods, such as Partial Dependence Plots (PDP), to contextualize the novelty and advantages of ALE.
3. It is recommended to change the vertical axis of Figure 1 from Frequency to Numbers; in addition, it is recommended to add subtitles to Figures 3/4/5, for example, Fig3a, b.
4. Some minor typographical errors were noticed throughout the manuscript (e.g., missing punctuation mark, break in third paragraph of page 2).
Comments on the Quality of English Language
Minor editing of English language required.
Round 2
Reviewer 1 Report
Comments and Suggestions for Authors
The authors have made the necessary corrections, and the article can be published in the journal in its current form.
Comments on the Quality of English LanguageMinor editing of English language required.
Reviewer 3 Report
Comments and Suggestions for Authors
In the revised manuscript, the authors have addressed all of my comments. I think the manuscript can be accepted in this version.